# Drug–Drug Interaction between Tacrolimus and Vonoprazan in Kidney Transplant Recipients

**DOI:** 10.3390/jcm10173964

**Published:** 2021-08-31

**Authors:** Yoshiharu Suzuki, Takuya Yoshihashi, Kazuhiro Takahashi, Kinji Furuya, Nobuhiro Ohkohchi, Tatsuya Oda, Masato Homma

**Affiliations:** 1Department of Pharmacy, University of Tsukuba Hospital, Ibaraki 305-8576, Japan; ysuzu-tuk@umin.ac.jp (Y.S.); tyoshihashi@times.hosp.tsukuba.ac.jp (T.Y.); 2Department of Gastroenterological and Hepato-Biliary-Pancreatic Surgery, and Organ Transplantation, Faculty of Medicine, University of Tsukuba, Ibaraki 305-8575, Japan; kazu1123@md.tsukuba.ac.jp (K.T.); kfuruya@md.tsukuba.ac.jp (K.F.); nokochi@mitochuo-hsp.or.jp (N.O.); tatoda@md.tsukuba.ac.jp (T.O.); 3Department of Pharmaceutical Sciences, Faculty of Medicine, University of Tsukuba, Ibaraki 305-8575, Japan

**Keywords:** tacrolimus, vonoprazan, kidney transplantation, drug–drug interaction, gene polymorphism

## Abstract

Kidney transplant recipients with tacrolimus-based immunosuppressive therapy are often treated with proton-pump inhibitors (PPIs) to prevent gastric ulcer complications. Vonoprazan, a potassium-competitive acid blocker, is a novel PPI possessing different metabolic pathways from conventional PPIs (e.g., omeprazole, lansoprazole and rabeprazole). However, no data are available on the change in blood concentration of tacrolimus after switching rabeprazole, a conventional PPI, to vonoprazan coadministration in the initial period of post-transplantation. This is a retrospective study of 18 kidney transplant recipients. The blood concentration and the concentration to dose (C/D) ratio of tacrolimus were compared before and after switching from rabeprazole to vonoprazan. Impacts of *CYP2C19* and *CYP3A5* genetic polymorphisms on the drug–drug interaction were also examined. The median (range) trough concentration of tacrolimus was significantly increased from 5.2 (3.6–7.4) to 8.1 (6.1–11.7) ng/mL (*p* < 0.0005) after switching from rabeprazole to vonoprazan. The C/D ratio of tacrolimus was also significantly increased from 38.1 (16.5–138.1) to 48.9 (26.2–207.2) (*p* < 0.0005). The percent changes of tacrolimus concentrations and C/D were 65.8% and 41.8%, respectively. *CYP2C19* and *CYP3A5* genetic polymorphisms did not affect the change in concentration and C/D ratio of tacrolimus. The present study indicates that vonoprazan coadministration increases the tacrolimus concentration regardless of *CYP2C19* or *CYP3A5* genetic polymorphisms. Thus, frequent monitoring of blood tacrolimus concentration is required when vonoprazan is introduced as an intensive gastric acid blocker in the early phase of post-transplantation.

## 1. Introduction

Tacrolimus, a calcineurin inhibitor, is used as an immunosuppressive agent for allograft transplantation in combination with mycophenolate mofetil and corticosteroids [1]. Tacrolimus requires frequent measurements of the blood concentration due to the narrow therapeutic range of 5–15 ng/mL [2]. Exceeding the therapeutic range induces tacrolimus-related toxicities (e.g., hyperglycemia, kidney injury, and infectious diseases), and subtherapeutic levels cause rejection episodes and graft-versus-host diseases in transplant recipients [3,4]. Therefore, conducting therapeutic drug monitoring of tacrolimus is important for optimizing the dose keeping the therapeutic range of the blood concentration in individual patients [5,6,7,8].

Since tacrolimus is mainly metabolized by cytochrome P450 (CYP) 3A4/5 [9,10], the concomitant use of drugs possessing CYP3A4/5 modification activity (inducers, inhibitors, and substrates) provides drug–drug interactions (DDIs). The conventional proton-pump inhibitors (PPIs), lansoprazole and omeprazole, which are substrates for CYP2C19 and 3A4, are often given with tacrolimus for preventing gastrointestinal complications in kidney transplant recipients [11]. Coadministration of these PPIs with tacrolimus has been known to produce DDIs where blood tacrolimus was increased, especially in patients carrying *CYP2C19*2* and **3* [12,13]. Another conventional PPI, rabeprazole, of which primary metabolic pathway is nonenzymatic, did not produce DDIs with tacrolimus [13,14,15,16,17].

Vonoprazan (TAK-438), a novel potassium-competitive acid blocker, has been approved for ethical use in Japan and several Asian countries. Vonoprazan provides a rapid onset of the action and potent suppression of gastric acid secretion compared with conventional PPIs [18,19]. It has been reported that the use of vonoprazan is advantageous as an alternative PPI for erosive esophagitis, *Helicobacter pylori* eradication, and gastric or duodenal ulcers [20,21,22]. Therefore, it is preferred for use in kidney transplant recipients who require intensive and rapid acid suppression for gastric symptoms with tacrolimus-based immunosuppression including corticosteroids. Coadministration with tacrolimus potentially produces DDIs because vonoprazan shares CYP3A4/5 with tacrolimus in its hepatic metabolism [23], though the clinical impact of DDIs is unclear.

The present study investigated the change in the blood concentrations of tacrolimus when PPI was switched from rabeprazole to vonoprazan in kidney transplant recipients with tacrolimus-based immunosuppression in terms of assessing the DDIs. The effects of genetic polymorphisms for *CYP2C19* and *CYP3A5*, responsible for conventional PPIs and tacrolimus metabolism, on these DDIs was also examined.

## 2. Materials and Methods

### 2.1. Patients and Drug Administrations

Eighteen kidney transplant recipients were recruited for this study from February 2015 to July 2020 (Table 1). The recipients were treated with tacrolimus-based immunosuppression, tacrolimus QD (Graceptor^®^; Astellas Pharma, Tokyo, Japan) in combination with mycophenolate mofetil and corticosteroids. The initial dose of tacrolimus QD was 0.15–0.20 mg/kg/day for 3–7 days before transplantation. The patients’ PPIs were switched from oral rabeprazole to vonoprazan (TAK-438; Takecab^®^; Takeda Pharmaceutical Company Limited., Tokyo, Japan) for intensive gastric acid suppression on postoperative days (POD) 3–43. The median (range) dose of oral rabeprazole and vonoprazan were 0.16 (0.11–0.22) and 0.31 (0.11–0.44) mg/kg/day, respectively (Table 1). Blood concentrations of tacrolimus were stable when the rabeprazole was switched to vonoprazan. Liver function for the patients was normal. Patients did not have any symptoms (e.g., nausea, vomiting, constipation, or diarrhea) after switching to oral PPIs. Major symptoms due to gastric ulcer, retching, and heartburn disappeared by switching from rabeprazole to vonoprazan. 

The study was approved by the ethics committee of the University of Tsukuba Hospital (approved number: H24–145). Written informed consent was obtained from all patients.

### 2.2. Determination of Blood Tacrolimus Concentration

Whole blood concentrations of tacrolimus were measured using the ARCHITECT chemiluminescent immunoassay system (Abbot, Abbot Park, IL, USA). Blood samples were collected in the morning just before the morning administration of tacrolimus to determine the trough levels. Tacrolimus dose was adjusted to achieve predefined target concentration ranges of 5–10 ng/mL (POD −3–14), 7–8 ng/mL (POD 14–28), and 5–8 ng/mL (after POD 28). The change in concentration/dose (C/D; in nanogram per milliliter/milligram per kilogram) ratio of tacrolimus before and after initiating oral vonoprazan was calculated to assess the DDIs.

### 2.3. Genomic DNA Extraction and Genotyping

Genomic DNA was extracted from peripheral whole blood samples using the QIAamp DNA blood kit system (Qiagen, Hilden, Germany) according to the manufacturer’s instruction. Subjects were genotyped for *CYP2C19* rs4244285 (*CYP2C19*2*), *CYP2C19* rs4986893 (*CYP2C19*3*), and *CYP3A5* rs776746 (*CYP3A5*3*) polymorphisms. These variants were identified by real-time polymerase chain reaction (PCR) using a TaqMan^®^ Drug Metabolism Genotyping Assays (Thermo Fisher Scientific, Waltham, MA, USA) with a QuantStudio 3 real-time PCR system (Thermo Fisher Scientific). Recipients were categorized into two groups based on the *CYP2C19* genotypes (extensive metabolizer (EM; *CYP2C19*1/*1*)), intermediate metabolizer (IM; *CYP2C19*1/*2* and **1/*3*), and poor metabolizer (PM; **2/*2*, **2/*3*, and **3/*3*)] and the *CYP3A5* genotypes (*CYP3A5*1* carrier (*CYP3A5*1/*1* and *CYP3A5*1/*3*)) and the noncarrier (*CYP3A5*3/*3*)].

### 2.4. Statistical Analysis

The Wilcoxon signed-rank test was used to compare the difference in the concentration and C/D ratio of tacrolimus before and after switching to vonoprazan. Differences in the concentration and C/D ratio of tacrolimus among the genotypes were compared by the Mann–Whitney U test. The statistical analyses were carried out using SPSS 26.0 (IBM, Tokyo, Japan). Results with a *p* value of <0.05 were considered statistically significant.

## 3. Results

### 3.1. Case Presentation

Typical change in the blood tacrolimus before and after switching the PPI from rabeprazole to vonoprazan are presented in Figure 1. Rabeprazole or vonoprazan was administered orally once a day after breakfast, and tacrolimus QD was given once a day 2 h after PPI administration. Trough tacrolimus concentrations increased from 3.8–4.7 to 9.4–10.7 ng/mL after switching from rabeprazole to vonoprazan. A similar increase was observed in the C/D ratio of tacrolimus from 26.2–29.0 to 52.1–65.3 after the switch. On the vonoprazan phase, a dose escalation from 10 to 20 mg provided a further increase in the C/D ratio from 33.6–37.1 to 41.3–48.8 (Figure 1). No difference was observed in the kidney and liver function before and after switching from rabeprazole to vonoprazan. Other concomitant medications remained unchanged during the observation period.

### 3.2. Effect of Vonoprazan Coadministration on the C/D Ratio of Tacrolimus

The blood concentration of tacrolimus increased after switching from rabeprazole to vonoprazan in 18 kidney transplant recipients (Table 1; Figure 2). The tacrolimus concentration (median (range)) after initiating vonoprazan was significantly higher than that before switching (8.1 (6.1–11.7) vs. 5.2 (3.6–7.4) ng/mL; *p* < 0.0005). The C/D ratio of tacrolimus 7 days after initiating vonoprazan was significantly higher than that before switching (48.9 (26.6–207.2) vs. 38.1 (16.5–138.1) (ng/mL)/(mg/kg); *p* < 0.0005). The percent changes in tacrolimus concentrations and the C/D ratio after the switching were 65.8 (0.0–192.5)% and 41.8 (2.7–129.8)%, respectively. No adverse event was observed in all patients because blood concentrations of tacrolimus after switching to vonoprazan were maintained in the therapeutic range. No difference was observed in the laboratory data of liver (aspartate aminotransferase and alanine aminotransferase) and kidney function (blood urine nitrogen and serum creatinine) between, before and after the switch. 

The effects of vonoprazan on tacrolimus concentrations and C/D ratio were compared between the subjects with and without *CYP2C19* or *CYP3A5* mutant alleles (i.e., *CYP2C19* EM vs. IM/PM or *CYP3A5*1* carrier vs. noncarrier). No significant differences were noted in increases in concentration and C/D ratio of tacrolimus after switching from rabeprazole to vonoprazan coadministration among patients with *CYP2C19* and *CYP3A5* genotype status (Table 2).

## 4. Discussion

A peptic ulcer is a common complication for kidney transplant recipients under immunosuppressive therapy including corticosteroids [24]. To prevent the peptic ulcer and related symptoms, PPI is commonly used in transplant recipients as a gastric acid blocker. Conventional PPIs, omeprazole and lansoprazole, produce DDIs with tacrolimus, where blood tacrolimus concentration was increased by competitively inhibiting tacrolimus metabolism on hepatic CYP3A4. Since omeprazole and lansoprazole are metabolized by CYP3A4 and CYP2C19 on their hepatic elimination, DDIs with tacrolimus occur especially in *CYP2C19* IM/PM patients whose blood concentration of PPIs are higher compared with *CYP2C19* EM patients [12,13]. Another conventional PPI, rabeprazole, whose main metabolic pathway is nonenzymatic, does not provide DDIs with tacrolimus [12,13]. Rabeprazole, therefore, is preferred as an alternative PPI for patients with tacrolimus-based immunosuppression [12,25,26]. Current interests have been focused on a novel PPI, vonoprazan, which acts as a potassium-competitive acid blocker and provides rapid and strong suppression on gastric acid secretion [18,19]. The use of vonoprazan has been recently introduced for kidney transplantation, especially in recipients with gastric symptoms resisting conventional PPIs. Whether or not vonoprazan produces DDIs with tacrolimus is important because both drugs are metabolized by CYPs for their hepatic elimination [9,10,23].

Blood tacrolimus concentration and the C/D ratio increased after switching PPI from rabeprazole to vonoprazan (Figure 2). Vonopranan coadministration produced a 1.7 and 1.4-fold elevation in blood concentrations and C/D ratio of tacrolimus, respectively (Figure 2). No differences in kidney and liver functions were noted before and after switching PPI (Table 1) and other concomitant medications were unaltered during the study periods. Therefore, vonopranan is considered to produce DDIs with tacrolimus in kidney transplant recipients under tacrolimus-based immunosuppressive therapy. A similar result has been reported in another study in which the C/D ratio of tacrolimus increased 1.1-fold after switching rabeprazole to vonoprazan in outpatients 2–7 years post-transplantation [27].

The inhibitory effects on CYP3A4/5 activity have been confirmed in in vitro experiments of the DDIs for vonoprazan [23]. A single dose of vonoprazan coadministered with clarithromycin, a well-known substrate and CYP3A4 inhibitor, resulted in an increase in the plasma concentration of clarithromycin in healthy subjects [28]. This information supports the present findings that vonoprazan coadministration produced DDIs with tacrolimus on CYP3A4 in their hepatic elimination.

The interindividual variability of the DDIs between tacrolimus and conventional PPIs, lansoprazole or omeprazole, is well-known to be associated with the *CYP2C19* genotype [12,25,26]. Since vonoprazan is also partially metabolized by *CYP2C19* in vitro [23], we examined the association between DDIs and *CYP2C19* gene polymorphism. No difference was found in the magnitude of DDIs with tacrolimus between *CYP2C19* EM and *CYP2C19* IM/PM (Table 2). This means that the *CYP2C19* genotype does not affect the vonoprazan pharmacokinetics [19] and the magnitude of DDIs between vonoprazan and tacrolimus. Vonoprazan was also reported to inhibit CYP3A5 activity as well as CYP3A4 [23]. The association between the DDIs and *CYP3A5* genotype status was investigated because the pharmacokinetics of tacrolimus is sensitive to *CYP3A5* gene polymorphisms [29]. Vonoprazan coadministration showed a slightly higher percentage increase of the C/D ratio in *CYP3A5*1*-carriers compared with noncarriers (44.4% vs. 39.1%), though not statistically significant. A similar result was reported in another study employing outpatients where switching to vonoprazan provided no significant increase of tacrolimus trough levels, even in the *CYP2C19* IM/PM in the *CYP3A5*1* noncarriers [30]. 

P-glycoprotein, a drug efflux transporter expressed in the intestinal lumen [31], can be considered in addition to CYP as the mechanism of the DDIs between vonoprazan and tacrolimus. Both rabeprazole and vonoprazan inhibit the transporter activity of P-glycoprotein with an IC_50_ of ≥100 and 50 µM [32,33]. According to the DDI guidelines [34,35,36], the maximum concentrations for rabeprazole and vonoprazan in the intestinal lumen are estimated to be 111 and 231 µM, respectively, when the drugs are administered at 10 and 20 mg, respectively. Since the intestinal concentration of vonoprazan is five times higher than the IC_50_ for P-glycoprotein inhibition, P-glycoprotein inhibition may participate in the mechanism of DDIs between vonoprazan and tacrolimus, at least theoretically. Vonoprazan coadministration may enhance the gastrointestinal absorption of tacrolimus via inhibiting P-glycoprotein, playing an important role in tacrolimus efflux at the intestinal lumen [37]. 

This study has several limitations. First, the study is a small retrospective one performed at single center, and the enrolled patients are in the early phase of post-transplantation. This may be a potential bias for the study setting. The increase in C/D ratio of tacrolimus after switching of rabeprazole to vonoprazan was large (41.8%) compared with previous studies by Mei T. [27] and Watari S. [30], which showed a 10.6% increase and no increase after the conversion, respectively, in the outpatient phase of post-transplantation. To confirm this difference, the data for a larger number of the patients is required in the early phase of post-transplantation. Secondly, the DDIs were assessed by the trough concentration of tacrolimus as the practical marker, instead of the area under the curve (AUC), because it had been reported that the trough concentrations showed good correlation with AUC in tacrolimus pharmacokinetics [38]. Since the use of AUC is a more precise way to assess tacrolimus DDIs, the change in AUC after conversion to vonoprazan should be confirmed to measure the magnitude of DDIs accurately in a future study.

## 5. Conclusions

In conclusion, the coadministration of vonoprazan provides DDIs with tacrolimus in kidney transplant recipients under tacrolimus-based immunosuppressive therapy. These DDIs may occur in CYP3A4 metabolism for their hepatic elimination, and are different from the conventional PPIs, lansoprazole and omeprazole, in terms of independence from the patient’s *CYP2C19* genotype. Since the change in blood concentration of tacrolimus ranged from 0% to 192.5%, a dose adjustment of tacrolimus is required according to close monitoring of blood concentrations when vonoprazan is concomitantly introduced in the early phase of post-transplantation.

## Figures and Tables

**Figure 1 jcm-10-03964-f001:**
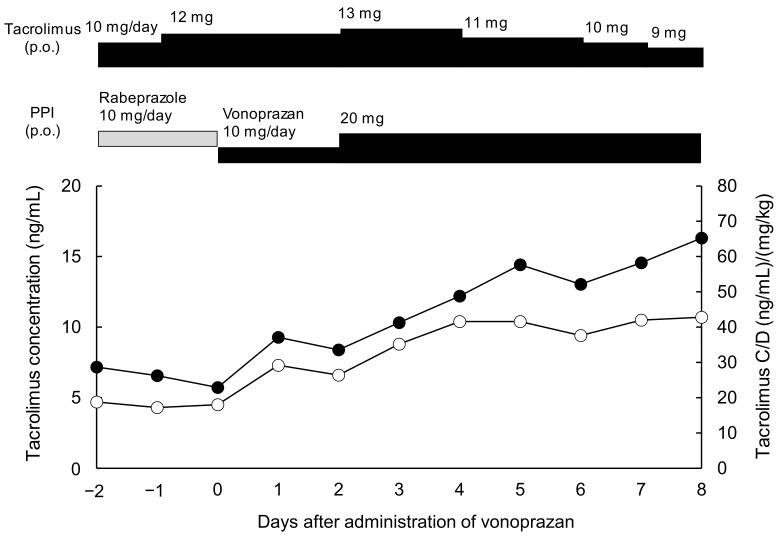
Changes in tacrolimus concentration (open circle) and C/D (closed circle) in a typical case switched from rabeprazole to vonoprazan. PPI, proton-pump inhibitor; C/D, concentration/dose; p.o., per oral.

**Figure 2 jcm-10-03964-f002:**
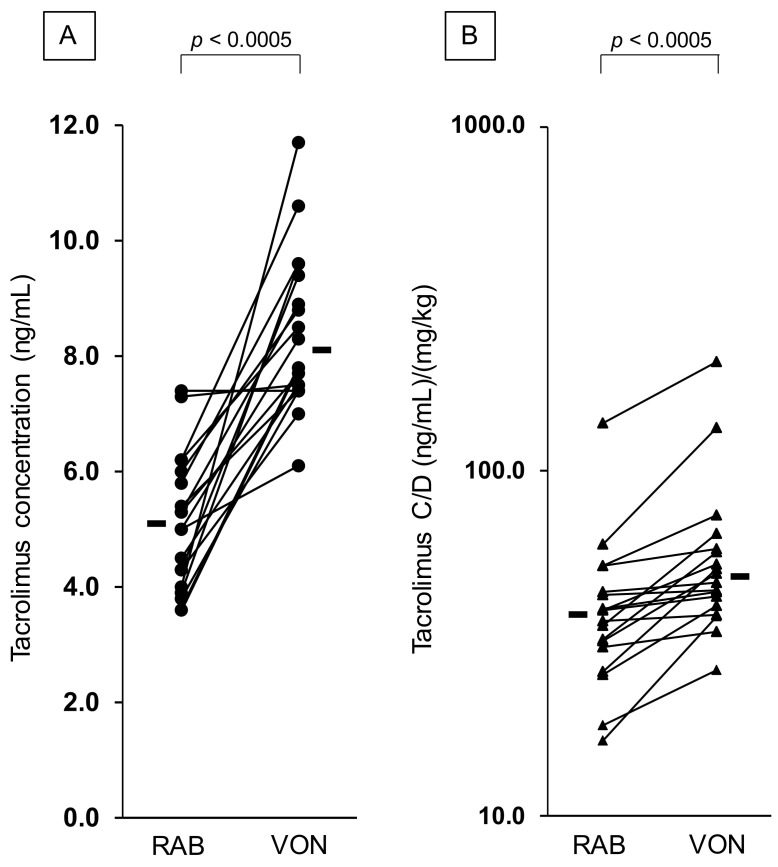
Effect of switching proton-pump inhibitor from rabeptazole to vonoprazan on the concentration (**A**) and C/D ratio (**B**) of tacrolimus. RAB, rabeprazole; VON, vonoprazan; C/D, concentration/dose. Bars indicate median values.

**Table 1 jcm-10-03964-t001:** Patient characteristics.

	Proton-Pump Inhibitor	
	Rabeprazole	Vonoprazan
Number of patients (male/female)	18 (12/6)	
Age (years)	45 (23–65)	
Body weight (kg)	61.9 (45.5–88.8)	
Tacrolimus		
Dose (mg/kg/day)	0.14 (0.05–0.24)	0.17 (0.04–0.31)
Concentration (ng/mL)	5.2 (3.6–7.4)	8.1 (6.1–11.7)
C/D ratio (ng/mL)/(mg/kg)	38.1 (16.5–138.1)	48.9 (26.6–207.2)
Laboratory data		
Aspartate aminotransferase (IU/mL)	16 (9–44)	15 (6–31)
Alanine aminotransferase (IU/mL)	28 (5–89)	22 (5–104)
Blood urine nitrogen (mg/dL)	24.2 (9.8–78.7)	17.4 (7.5–45.4)
Serum creatinine (mg/dL)	1.55 (0.62–4.31)	1.21 (0.58–2.84)
Proton-pump inhibitor		
Dose (mg/kg/day)	0.16 (0.11–0.22)	0.31 (0.11–0.44)
Administration period (days)	9 (1–37)	7 (5–13)

Data are presented as number or median (range). C/D, concentration/dose ratio.

**Table 2 jcm-10-03964-t002:** Comparison of tacrolimus trough concentration and the C/D ratio among the various *CYP* genotypes.

		Tacrolimus Concentration (ng/mL)						Tacrolims C/D Ratio [(ng/mL)/(mg/kg/day)]					
Genotype	n	Rabeprazole		Vonoprazan		ΔConcentration		Rabeprazole		Vonoprazan		ΔC/D Ratio	
*CYP2C19*													
EM	5	4.0	(3.6–6.2)	8.5	(7.4–11.7)	3.6	(2.3–7.7)	25.6	(16.5–53.5)	40.7	(26.6–74.4)	15.1	(8.2–21.5)
IM/PM	13	5.3	(3.9–7.4)	7.7	(6.1–10.6)	3.0	(0.0–5.7)	39.5	(26.2–138.1)	50.4	(34.3–207.2)	6.3	(1.2–73.3)
*CYP3A5*													
**1* carrier	5	5.3	(3.6–7.4)	8.5	(7.0–11.7)	2.8	(0.0–7.7)	36.9	(16.5–138.1)	45.1	(26.6–207.2)	14.6	(1.2–69.1)
**1* noncarrier	13	5.0	(3.8–6.2)	7.5	(6.1–10.6)	3.3	(1.1–4.4)	44.9	(30.8–61.2)	50.4	(34.3–134.5)	18.3	(2.5–73.3)

Data are shown as number or median (range). EM: Extensive metabolizer, *CYP2C19*1/*1*; IM: Intermediate metabolizer, *CYP2C19*1/*2, *1/*3*; PM: Poor metabolizer, *CYP2C19*2/*2*, **3/*3*, **2/*3*; **1* carrier, *CYP3A5*1/*1*, **1/*3*; **1* noncarrier, *CYP3A5*3/*3*.

## Data Availability

The data that support the findings of this study are available from the first author, Y.S., and the corresponding author, M.H., upon reasonable request.

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
