# Peer review of "Drug–Drug Interaction between Tacrolimus and Vonoprazan in Kidney Transplant Recipients"

_jcm, 2021, doi:10.3390/jcm10173964_

Round 1
Reviewer 1 Report
The pharmacokinetics of drug interactions between immunosuppressants and proton pump inhibitors is an important element of research into the efficacy of immunosuppressive therapy. The emergence of new PPI inhibitors makes it necessary to conduct in-depth research on their pharmacokinetics, genetic background related to polymorphisms and interactions with immunosuppressants. The authors of the publication assessed all these considerations in the context of wonoprazan, which acts as a potassium competitive acid blocker and provides rapid and strong suppression of gastric acid secretion. They rightly noticed that wonoprazan may interact with tacrolimus in the stage of hepatic metabolism by cytochromes, and thus may be subject to polymorphic variability. As it turned out, they obtained interesting results concerning the interaction of both of these drugs, their influence on kidney and liver function, and concerning the concentration of other drugs administered simultaneously. The most interesting seems to be the lack of influence of the CYP2C19 polymorphism in CYP2C19 EM and CYP2C19 IM / PM carriers on the pharmacokinetics of vonoprazan. Moreover, the authors made an interesting discovery regarding the genotype of CYP3A5 and P-glycoprotein in relation to the bioavailability of the specific drug - tacrolimus. The publication is interesting, it touches upon a very important aspect of pharmacotherapy and contributes to the improvement of the patient's pharmacotherapy.
Author Response
Reviewer 1:
Thank you for useful comments on our manuscript.

Reviewer 2 Report
The paper from Suzuki et al. is retrospectively analyzing the changes in tacrolimus trough concentration after early post-transplant switching from rabeprazole to vonoprazan treatment in 18 kidney transplant recipients.
The authors found a significant increase in tacrolimus trough levels and concentration/dose ratio after switching from rabeprazole to vonoprazan.
The topic is of interest and the English language is appropriate.
However the paper has several limitations:
1) the reference list should be updated with the recent papers published on this topic (i.e. Watari Drug Metabolism and Pharmacokinetics 2021;40:100407)
2) the previous published papers on bigger populations on this topic found no statistically significant difference in the tacrolimus trough levels, C/D ratio and AUC, even if they were conducted lately after transplant (transplant vintage of 42 months in the Watari paper and 59 months in the Mei paper). Can the authors comment on that?
3) the only original finding of this paper is the early post-transplant convertion time, however this can result in biases.
4) the authors are taking into considerations only trough levels and C/D ratio, without analyzing the AUC, this is essential for a pharmacokinetic study in the context of CYP450 inhibitors.
Author Response
Reviewer 2:
Thank you for useful comments on our manuscript. We revised the manuscript according to your comments as follows:
1) According your suggestion, we updated the references including currently published paper (Watari S., et.al., Drug Metab Pharmacokin, 2021) (reference No 30) and added the following explanation regarding the effects of genetic variation on the DDIs in discussion section as follows:
Similar result has been reported in other study employing outpatients where switching to vonoprazan provided no significant increase of tacrolimus trough levels, even in the CYP2C19 IM/PM in the CYP3A5*1 noncarriers [30]. (P7, line 231–233)
2) According your suggestion, we added the comments on the previous papers, which were superior in the patient number, as the study limitation in the discussion section. As you indicate, the difference between present study and previous one’s is the conversion time, early (inpatient period) or late (outpatient period) phase of post-transplantation. We, therefore, make this clear as the originality and study limitations in discussion section as follows:
This study has several limitations. First, present study is a small retrospective one performed at single center and the enrolled patients are in the early phase of post-transplantation. These may be potential bias for the study setting. The increase in C/D ratio of tacrolimus after switching of rabeprazole to vonoprazan was large (41.8%) compared with the previous studies, Mei T. [27] and Watari S. [30], which showed 10.6% increase and no increase after the conversion, respectively, in outpatient phase of post-transplantation. To confirm this difference, the data for large number of the patients will be required in early phase of post-transplantation. (P8, line 246–253).
3) According your suggestion, we clearly present our original findings in discussion section (P8, line 252–259, see above) and conclusion (as shown in below).
Since the change in blood concentration of tacrolimus ranged 0%–192.5%, the dose adjustment of tacrolimus is required according to close monitoring of blood concentrations when vonoprazan is concomitantly introduced in early phase of post-transplantation. (P8, line 264–267)
4) We agree with your comments concerning the use of AUC in full PK study. We, therefore, added the sentence indicating importance of AUC for assessing the magnitude of DDIs in discussion section as the study limitation.
Secondly, the DDIs was assessed by trough concentration of tacrolimus as the practical marker instead of the area under the curve (AUC), because it had been reported that the trough concentrations showed good correlation with AUC in tacrolimus pharmacokinetics [38]. Since the use of AUC is more precise way to assess tacrolimus DDIs, the change in AUC after conversion to vonoprazan should be confirmed to measure the magnitude of DDIs accurately in future study. (P8, line 253–258).
